# *PauseNørd* Pilot Study: Exploring the Implementation of Mini Movement Breaks in University Lectures

**DOI:** 10.3390/ijerph22050739

**Published:** 2025-05-07

**Authors:** Ilaria M. Piccinini, Jesper Hallas, Casey L. Peiris, Henrik Lauridsen, Tina Dalager, Karen Søgaard

**Affiliations:** 1Center for Muscle and Joint Health, Department of Sports Science and Clinical Biomechanics, University of Southern Denmark, 5230 Odense, Denmark; hlauridsen@health.sdu.dk (H.L.); tdalager@health.sdu.dk (T.D.); 2Clinical Pharmacology, Pharmacy and Environmental Medicine, Department of Public Health, University of Southern Denmark, 5230 Odense, Denmark; jhallas@health.sdu.dk; 3Pharmacology, Odense University Hospital, 5000 Odense, Denmark; 4School of Allied Health, Human Services and Sport, Physiotherapy, La Trobe University, Melbourne, VIC 3086, Australia; c.peiris@latrobe.edu.au; 5Allied Health, The Royal Melbourne Hospital, Parkville, VIC 3052, Australia; 6Department of Clinical Medicine, Aarhus University, 8200 Aarhus, Denmark

**Keywords:** classroom movement breaks, mind breaks, university students, attention, well-being

## Abstract

We pilot-tested movement breaks lasting < 1 min at the University of Southern Denmark (SDU) with an initiative called *PauseNørd*. Our mixed-methods feasibility study explored the acceptability, practicality, expansion, and limited efficacy of the *PauseNørd* breaks (PN breaks) via observation, questionnaires, and short interviews. Three groups of students tested lectures with and without the PN breaks. The PN breaks were well accepted by students and lecturers, and they were practically easy to implement. Suggestions to improve practicality related to timing and frequency. Potential barriers to expansion were identified in relation to student familiarity with exercise and lecturer comfort in leading the breaks. Preliminary data on effectiveness indicated that the PN breaks could support alertness, concentration, enjoyment, motivation, positive mood, and the social interactions within the class. The learning from the pilot study will be used to inform future PN research.

## 1. Introduction

Although difficult to measure, maintaining student attention and well-being during university lectures remains a persistent challenge for educators [1,2]. Long lectures and prolonged sitting have been associated with fatigue and reduced well-being among students [3,4], potentially impairing academic performance [5,6]. Incorporating brief “brain breaks” into lectures may help restore mental energy, with social and physical activities during these breaks being linked to increased vigour and improved well-being [7].

Classroom movement breaks were developed to be “brief exercise breaks designed to be time efficient and feasibly implemented in the classroom” [8]. They have been referred to as “classroom movement breaks”, “exercise breaks”, “brain breaks”, “short bouts of physical activity”, and “energizers”. Although research has been conducted mostly in primary and secondary schools [9,10,11], recent studies have started exploring the benefits of movement breaks among university students [12,13,14,15,16,17,18,19,20]. Movement breaks can positively influence on-task attention and memory [21], cognitive performance [22], and academic achievement [14]. Studies that explored movement breaks during university lectures indicate they are feasible, may increase attention and well-being, and reduce fatigue and sedentary behavior [13,17,20]. However, the duration of previous class-based studies in universities was generally from four to ten minutes, which may be disruptive to the delivery of teaching content in a university setting. Lab-based studies showed that one-minute relaxing and detaching activities could have an effect on the replenishment of the psychological resources lost during a working task [23,24]. In one study, 30 s stretching every 10 min did not reduce attention and increased job performance [25]. Brief exercise breaks have also been found to have cardiometabolic and other health benefits in adults [26,27]. We are not aware of any classroom-based study that has explored the feasibility and the potential benefits for attention and well-being of mini movement breaks with a maximum duration of one minute among university students.

For these reasons, we decided to explore the feasibility of an initiative born at the University of Southern Denmark (SDU) called *PauseNørd*. It consists of the integration of mini movement breaks lasting less than one minute within a lecture with the intention to provide a brain break. Initially introduced during the COVID-19 pandemic in an online learning course in pharmacology, it was later adapted to in-person lectures with movements or exercises that could be easily performed in a restricted space (see the history of *PauseNørd* in Appendix B). However, the feasibility and efficacy of these mini breaks has not been evaluated. Therefore, we performed a mixed-methods pilot study of the *PauseNørd* (PN) breaks to inform a full study.

Our first objective was to investigate four elements of the feasibility of integrating the PN breaks during university lectures (see the subsection on Outcome Measures). These were chosen among the domains identified by Bowen and colleagues [28], i.e.,

(i).Acceptability to students and lecturers;(ii).Practicality (time, frequency, and kind of lecture);(iii).Expansion (potential barriers and facilitators);(iv).Limited efficacy testing and comparison with studies with a similar design.

As no validated questionnaire was found to assess limited efficacy testing for our purposes, our second objective was to pre-test two kinds of questionnaires and two kinds of testing timing in different scenarios, assessing their feasibility–acceptability for future research.

## 2. Materials and Methods

### 2.1. Study Design and Ethics

The *PauseNørd* pilot study was a concurrent mixed-methods feasibility study to explore the acceptability, practicality, expansion, limited efficacy, and assessment of the PN breaks (less-than-one-minute mini movement breaks). Mixed methods were used to gain a more comprehensive understanding of the findings and included observation, a questionnaire for students, and interviews with both students and lecturers. The pilot study was conducted at the University of Southern Denmark from November to December 2023. In Denmark, projects of this kind do not require ethical consent (Danish national legislation “LBK nr 1268 from 28 December 2024”); however, students provided implied consent by participating in the voluntary movement breaks and completing the surveys.

### 2.2. Participants and Recruitment

The conditions for recruitment and data collection were limited to one month due to funding and schedule constraints. Students were eligible to participate if they were able to safely engage in short movement breaks and were attending lectures delivered by lecturers (n = 2) in the Department of Health who had agreed to deliver the PN breaks.

The study recruited via convenience sampling, and the methods were tailored to align with the participant groups that were available for the study. An introduction to the study was given by one of the researchers together with the lecturer prior to the first lecture for each group of students explaining the intervention and data collection. Participation was voluntary, and students provided implied consent if they chose to participate in the PN breaks and complete the questionnaire and/or interview. Two lecturers agreed to participate in the pilot study involving three groups of students in the Department of Health at SDU: (i) Group 1, master’s students in Pharmacy (n = 14), who had lectures based on the flipped-classroom principle, so that the students had to read some material prior to the class; (ii) Group 2, bachelor’s students in Medicine (n = 81), who had a classic lecture and participated for one hour only; (iii) Group 3, master’s students in Physiotherapy (n = 21), who alternated hours of classic lectures and group assignments, especially in Period II. All kinds of lectures included in this study (flipped, classic, and group assignment) shared the same schedule of alternating 45 min lectures (equivalent to 45 min sitting time) and 15 min official break intervals between lectures.

### 2.3. The PauseNørd Pilot Intervention

Intervention lectures included the integration of two PN breaks per 45 min class, approximately one every 15 min. There was no specific indication about the intensity of the activity. Control lectures did not include any break. A PN drawing represented the signal to stand up and perform a couple of physical exercises for a maximum duration of one minute, as led by the lecturer (Figure 1).

The exercises were chosen by the lecturer from the list provided (Box 1). When possible (Group 2 only), the PN drawings were placed within a slideshow. When not possible (Group 1 and 3), a signal at the right time was given by one of the researchers to the lecturer, who would then initiate the PN break. The scheduled 15 min interval after each lecture occurred as usual after both intervention and control lectures.

Box 1Sample of exercises used during the PN breaks.Examples of exercises/movements extracted from the list provided by the Sports Science and Clinical Biomechanics Department (IOB). The *PauseNørd* movement breaks were led by the lecturers and lasted less than one minute. The students could perform the exercises at their own desk/row. The goal was to provide a short mental break for students (and lecturers).
Squat

Core stretch with crossed legs

Frog movement exercise

Tree pose

Apple pick up stretch

Jumping Jacks

Egyptian mobility exercise

Palm tree movemen

Neck stretch



### 2.4. Outcome Measures

The primary outcome of this pilot study was the feasibility of the breaks. Feasibility was evaluated according to Bowen’s framework in terms of acceptability, practicality, expansion, and limited efficacy testing [28] (Table 1).

Acceptability refers to how the participants and lecturers reacted to the breaks [28] and was measured via observation of students (to determine participation and engagement) and via interviews with students and lecturers (to explore perceptions).

Practicality relates to how well an intervention can be delivered in a real-world situation when resources, time, or commitment might be limited or constrained in some way [28]. We measured practicality via direct observation (to assess practicality aspects related to space, timing, frequency, and safety) and interviews with students and lecturers to explore perceptions and suggestions to improve practicality.

The outcome of expansion refers to the ability of a successful intervention to be delivered to a different population or in a different setting [28]. This was assessed via interviews with students and lecturers to explore barriers and facilitators for future use of movement breaks.

Limited efficacy testing in feasibility studies relates to testing conducted in a convenience sample, with intermediate rather than final outcomes, with shorter follow-up periods, or with limited statistical power [28]. We assessed the efficacy of the breaks in relation to alertness, concentration, enjoyment, mood, sleepiness, cognitive fatigue, and restlessness via student questionnaires. These factors were chosen as they have been previously explored in studies related to students breaks [5,19,20,21,24,29].

The secondary outcome was the acceptability of our two questionnaires and testing frequency/timing, which was assessed by observation and interview of both students and lecturers.

### 2.5. Data Collection

Data collection occurred in the fall semester of 2023 and lasted from mid-November to mid-December, in the last part of the semester before the exams. An overview of the research process is illustrated in Figure 2.

*Qualitative data* were collected via observation and interview. Observation occurred in all periods and was performed by one of the researchers (IMP). IMP witnessed as a passive participant observer how the PN breaks were integrated and the overall atmosphere during the study time (body language, participation levels, and attentiveness) and engaged in informal talks with some students during the normal intervals. Students who had participated at least once in the PN breaks were invited to a focus group. However, as no student replied to the invitation, the students were asked if they wanted to answer a few questions for research purposes related to their experience with the PN breaks during the official 15 min interval between classes in Periods II and III. The semi-structured focus group interview was then adapted for the short field interviews to collect experiences and opinions (text in the Appendix A). Lecturers were contacted by e-mail to agree on a convenient time for a one-to-one interview.

During the interviews, voices were recorded to capture opinions accurately and were deleted once the text was transcribed. Before the interview, students were informed that personal data were used for research purposes according to the Danish Data Protection Act §10 and the General Data Protection Regulation art. 6 (1)(e). A copy of the information on the processing of personal data was given to the interviewee.

*Quantitative data* were collected via questionnaires and observation during the lectures. The study used a cross-over approach, so that the same group of students answered the questionnaires both during lecture hours with the PN breaks and during hours without the PN breaks.

The questionnaire chosen was based on previous research [20] and provided in the English language with definitions to make sure the students understood the concepts. Master’s students in SDU have some courses taught in the English language and are therefore proficient in this language. The questionnaire provided quantitative data on 9 items on a Visual Analogue Scale (VAS) related to characteristics that may affect attention and well-being within a lecture. We chose a conceptual framework of a reflective model [30] and asked participants to self-assess alertness, concentration, enjoyment related to the class, motivation for following the class, social interactions within the class, and positive mood as positive-valence variables, and sleepiness, cognitive fatigue, and restlessness due to prolonged sitting as negative-valence variables (full text of the questionnaire in the Appendix A).

The pilot aimed to test different methodologies related to the timing and kind of questionnaires (Figure 2). In most cases, the Q1 questionnaire was used at the start and end of the 45 min lecture. The Q1 questionnaire asked students about the 9 items (sentiments and cognitive states) in the moment. With questionnaire Q1, two timing methodologies were tested, which resulted in 2 questionnaires per 45 min lecture:Timing 1 (T1): measurements were taken at the start and end of the lecture, within 5 min from the beginning and the end of each lecture (e.g., at minutes 4 and 41).Timing 2 (T2): measurements were taken before and after the PN breaks, within 5 min time from the two PN breaks (e.g., if the PN breaks were at minutes 15 and 30, measurements would be at minutes 11 and 34).

One group was given questionnaire Q2, which asked them to summarize sentiments and cognitive states throughout the class and was administered at the end of each lecture (i.e., one questionnaire per 45 min lecture) (timing 3, T3).

Except for Group 1 in Period I, who completed a paper-based questionnaire with a unique identifier (Scenario 1), electronic questionnaires were distributed via SurveyXact (using a link/QR code). In this latter case, unique identifiers were not available, so that changes over time only could be observed on the group level. The test of different timings and the questionnaires created seven different pilot-study scenarios (Figure 2).

The timing and frequency of the breaks and the questionnaires and participation rates were collected through an observation checklist during all tested hours (IMP).

### 2.6. Data Analysis

Quantitative data were analyzed with Stata18. Data were analyzed separately for each scenario. For all variables, score means and differences in score means between lectures with and without the PN breaks are reported. However, as within-individual changes were not available, SD and 95% CI values are not reported. A Wilcoxon signed rank test was performed in Scenario 1. Two observations from Group 2 were excluded, one at baseline and one at endline, as the questionnaires reported several missing scores, while the few self-reported scores were all zeros.

Qualitative data: observation and interview data were combined in the analysis to validate observed behaviors with the students’ own reflections. The results were presented separately to distinguish between recorded behaviors and self-reported experiences. Transcriptions of the interviews were transferred to NVivo 14 software and analyzed following content analysis [31]. After a preparation phase and familiarization with the data, a deductive approach was chosen. Data were organized per subcategory and finally summarized per area of feasibility researched (main categories).

Integration of qualitative and quantitative data for limited efficacy testing occurred during the interpretation of the findings: when available, quantitative data were analyzed first. Qualitative data provided a context and deeper insights into the reasons and the preferences of the participants.

## 3. Results

As a new non-validated questionnaire was used, an exploratory factor analysis (EFA) was considered meaningful to provide insights into the structural validity of the questionnaire and the internal consistency of the variables, while allowing us to reduce the items to a few meaningful constructs. Details regarding the EFA can be found in Appendix C. From the EFA results, the following summarizing constructs are presented in this section:(i).“Engagement facilitators” (including alertness, concentration, enjoyment, motivation, and positive mood);(ii).“Engagement barriers” (including cognitive fatigue and restlessness).(iii).“Social interactions” and “sleepiness” were used as single variables.

### 3.1. Research Question 1: Feasibility of the PN Breaks

#### 3.1.1. What Is the Acceptability of the PN Breaks?

*Observation data*. The participation rate for the PN breaks was close to 100% and was consistently very high over time, with only one or two students occasionally not taking part (Table 2). Timing for both the lectures, the PN breaks, and the questionnaires was followed quite precisely in most cases. The atmosphere was positive, and the students who took part seemed to enjoy the movement breaks. Students in Pharmacy and in Medicine followed the exercises as led by their lecturer, who included two or three different movements per PN break, varying the kind of exercise almost every time. Compared to the other groups, the physiotherapy students engaged more in intensity and precision while doing the exercises. As they were professionals, the lecturer told them they could choose which exercise to do, so some of them selected their own. In all groups, it was noticed that they passed in a quick and ordered manner from sitting and engaging in the lecture to the PN exercises, and vice versa. One student from Group 3 casually mentioned she would have liked longer exercise breaks. No accidents were observed or reported. During the official 15 min intervals between classes (i.e., not the PN movement break), not all students got up. Some students remained seated, watching their phones or talking to other students.

*Interview data* (Table 3, row 1). The interviewees were (i) four students from the Master’s in Pharmacy course, (ii) three students from the Master’s in Physiotherapy course, and (iii) the two lecturers who agreed to participate in the intervention. Both students and lecturers said they enjoyed the PN break experience. In one case, one student suggested it is better suited for courses that are not as hard as a master’s; however, most of the students expressed that it was better to have more breaks if the content of a lecture was hard, as a mental break was then more needed.

#### 3.1.2. What Is the Practicality of the PN Breaks?

*Observation data.* The PN breaks were easy to integrate in a lecture, especially when the PN drawing could be inserted in the slideshow. Opening a file with the sole purpose of viewing the image was time-consuming. That step was skipped in Group 3, allowing an easier process.

*Interview data* (Table 3, row 2). Except for one student, who said he would always be ready to make some physical activity, most students had suggestions to improve practicality related to timing, frequency, the kind of lecture, the difficulty level of the lecture, etc. In general, it was considered easy to integrate and not disruptive in most cases. In one case, a student decided not to take part because she was not feeling well.

No difficulty was mentioned related to performing the exercises in a restricted space. No comments were made on the kind of exercises. Sometimes the frequency of the breaks was perceived as high for the specific kind of lecture/time, but overall they were found to be beneficial.

#### 3.1.3. What Is the Possibility of Expanding the PN Breaks in Other University Contexts?

*Interview data* (Table 3, row 3). Some barriers were identified related to the possibility of expanding the intervention to all lectures. Some of the students suggested they could imagine the PN breaks being well integrated in all kinds of classic lectures, especially the less interactive ones. The lecturers found that PN could be potentially integrated in all contexts but identified the background/culture of the students and lecturers as an important barrier. All students and lecturers involved were part of the Health Faculty, and the Physiotherapy group was specifically open to movement and aware of the benefits of physical activity. One student in Physiotherapy also shared this comment, underlining that they are a very special population. The lecturers also insisted on the fact that lecturers should feel comfortable in the teaching and leading the movements.

#### 3.1.4. What Are the Limited Efficacy Testing/Preliminary Data on the Effectiveness of the PN Breaks? And How Do the Results Compare with Findings from Other Studies That Used Similar Methods?

Group 2 could be tested for one lecture only (with PN breaks); however, the available results are reported for transparency and to give a partial insight.

*Interview data* (Table 3, row 4). Students and lecturers found that PN was a short and collective movement break experience, a mental break which could potentially improve the attention, the energy, the mood, and the social interactions in the class. Both lecturers mentioned that the PN breaks had some benefits on them as well.

*Questionnaire data*. Table 4 summarizes the data from the first two hours tested in Group 1 (Scenario 1). The majority of the students indicated a score increase in the positive-valence variables, 9 out of 10 for engagement facilitators (*p*-value = 0.0029) and 10 out of 11 for social interactions (*p*-value < 0.001). The negative-valence variables (engagement barriers and sleepiness) did not show statistically significant changes.

Table 5 describes the potential impact of PN on various scores across groups and timepoints (Scenarios 1, 2, 3, 4, 5, and 6). Positive-valence variables showed a score increase attributable to the PN breaks in all scenarios, except for a slight decrease of 0.3 in Scenario 4 for engagement facilitators. In half of the cases, the score increase was above 1 point on the VAS scale. Group 1 exhibited a higher increase: in Scenario 1, there was a score increase of 1.8 in engagement facilitators and an increase of 3.3 in social interactions. Engagement barriers exhibited a decrease in all scenarios, which was between −0.2 and −0.8. On the other side, sleepiness exhibited fluctuations between positive and negative values: there was an increase of between 0.1 and 0.5 in Group 1 and a decrease of between −0.7 and −0.5 in Group 3. In Group 2, only the changes during the hours with PN breaks were tested: the changes indicated an increase in all positive-valence variables and a decrease in the negative-valence variables.

Table 6 presents the score means and mean differences from the testing of questionnaire Q2 (Group 3, Scenario 7), which allowed us to compare the data with the study by Peiris et al. (2021), which used similar methods [20]. Except for social interactions, which displayed a negative mean difference of −0.7, positive-valence variables increased and negative-valence variables decreased with the PN breaks. Furthermore, alertness, concentration, motivation, cognitive fatigue, and restlessness exhibited a change above one point. The score means were similar to the ones found by Peiris and colleagues, except for enjoyment in the absence of movement breaks, which was 6.7 in the PN study and 5.5/5.6 in Peiris et al. [20].

### 3.2. Research Question 2: Feasibility of the Questionnaire/Testing Method

#### What Is the Acceptability of the Testing Method?

Participation rates for the questionnaires ranged from 100% to over 60%, showing higher rates in Period I and a decline in Periods II and III (Table 2). It was agreed with the lecturers that the observer would give the timing for the questionnaires and the breaks, except for Group 2, where all the material was inserted in the slideshow. This notification requested a short time for the lecturers to get organized with the slides to insert the link/QR code for the questionnaire and/or the PN break drawings. Although they took a fairly short time to complete (around one minute), it was noted that the questionnaire times increased with the number of interruptions and created more disruption in the lectures compared to the PN breaks (around two–three minutes per questionnaire), with the risk of giving survey fatigue. Questionnaire Q2 had the advantage of interrupting each lecture only once.

Some mentioned that the time dedicated to filling in the questionnaires was disruptive, although necessary. Lecturer 1, who had previous experience with the PN breaks, mentioned that it felt like a different and more disruptive experience.

## 4. Discussion

### 4.1. Summary of Findings

In our pilot study, the integration of the PN breaks (mini movement breaks lasting less than one minute) was well accepted by students and lecturers and was practically easy to implement. Suggestions to improve practicality were reported related to timing and frequency. Although acceptable to the students and lecturers in this pilot study, barriers to expansion in other university contexts were identified in the background/culture of the students and lecturers and the level of comfort that the lecturers may experience in leading the exercise breaks. Preliminary data on effectiveness indicated that the PN breaks could support the factors of positive engagement throughout a lecture (which included alertness, concentration, enjoyment, motivation, and positive mood) and the social interactions within the class. Moreover, the benefits related to both students and lecturers. However, there was no constant decrease in the negative-valence variables. The questionnaire Q2 with a QR code/link inserted within the slideshow was found to be the least disruptive method.

### 4.2. Acceptability, Practicality, and Expansion

The previous class-based studies which focused on university students found that classroom movement breaks were also feasible in terms of acceptability and practicality, although they all used different methods (duration, intensity, and frequency of the breaks, as well as the duration of the lectures) [15,19,20,32]. Keating et al. collected mostly positive responses from 111 students on acceptability, appropriateness, and feasibility for a 4 min active break [32]. Ferrer and Laughlin concluded that brain breaks are simple to implement and that lecturers can easily choose between a variety of activities and lengths depending on what is needed [15]. A standing and an active break of 5 min were compared to a free-choice open break in another study [19], which reported positive opinions of the lecturers involved. The tutors interviewed by Peiris et al. did not perceive that breaks impacted curriculum delivery [20]. A new study in 2024 also explored the feasibility of 5 min student-led exercises performed twice within 80 min in both smaller and larger courses, receiving positive feedback from the students [33]. We did not find other class-based studies assessing feasibility–expansion. This previous research supports our findings that brief movement breaks are broadly acceptable to both students and their lecturers.

However, lecturers in Paulus et al.’s study (2021) perceived that a 5 min break in a 90 min lecture was too long. And both students and tutors in Peiris et al. (2021) found that 5–10 min movement breaks every 20 min were too frequent and suggested that reducing the frequency or duration would increase practicality [19,20]. In our study, with a maximum of 1-minute breaks every 15 min, opinions about the frequency varied. Most of the students found that two short breaks were always feasible, non-disruptive, and beneficial; however, some students suggested it would be better to have more in the afternoon and in the more difficult classes (e.g., statistics) and fewer in the morning and in the more interactive classes.

### 4.3. Limited Efficacy Testing

In our pilot study, compared to no breaks, the positive-valence variables showed an increase, suggesting the PN breaks to be beneficial. This adds to the growing body of literature related to all ages and settings that indicates that short movement breaks are valuable [9,10,11,13,17,21,25]. Studies on breaks and micro-breaks mostly refer to the effort recovery model (ERM) as a theoretical framework to explain these results [34]. The ERM suggests that work tasks reduce employees’ energetic and cognitive resources and that breaks during the workday are necessary to re-establish those resources (“to recover”). Moreover, our study tested movement breaks lasting less than one minute, while other class-based studies in universities had previously tested breaks of four to ten minutes [17,18,19,20,33]. When compared to those reported by Peiris and colleagues, the results obtained in Scenario 7 (Table 6) suggest that the duration of less than one minute could be enough to restore alertness and concentration and improve enjoyment related to the class [20].

One important exception to the increase in the positive-valence variables is represented by Scenario 7, in which social interactions decreased. However, the hour tested without the PN break was mostly dedicated to group-work assignment, with high engagement between peers.

It was observed that negative-valence variables did not decrease in a stable way. One explanation could be that people may perceive a question asked in a positive or negative way differently [35]. Another possibility is that some students were confused by the change of direction (negative-valence versus positive-valence variables) in some of the questions.

### 4.4. Strengths and Limitations

To the best of our knowledge, this is the first study on mini movement breaks lasting less than one minute in the setting of a university lecture. Different groups and methodologies during the normal academic routine were tested to assess feasibility, which led to a variety of information. The presence of both quantitative and qualitative methods allowed us to have a more comprehensive view of the topic and helped mitigate the limits of using one method only, namely, the inability to capture the experience of the participants in quantitative data and the limited sample size of the qualitative data.

This pilot study has some obvious limitations. Blinding was not possible; however, the objectives of the study were not mentioned during the initial presentation to avoid influencing the participants. Observation, interviews, and qualitative data analysis were conducted by one researcher only (IMP). While observations were conducted systematically, the absence of validated tools may have affected the accuracy and replicability of the findings. The study was completed in a limited time period and adopted a convenience sample of three groups in the Faculty of Health, with the largest group being students in Medicine. Results may therefore not be generalizable to other populations. As there was variation in intervention delivery and testing among courses for pragmatic reasons, we analyzed the data separately for each scenario. This prevented us from using statistical modeling and taking other potential confounders into account. The number of participants and the number of testing hours were also limited. However, the overall results for positive-valence variables showed evidence of a positive change in Scenario 1 and are supported by the previous literature.

VAS scales are not specifically validated for attention and well-being in university students. However, a VAS scale is a known validated tool for the self-assessment of a variety of states and sentiments in several populations, e.g., pain [36], quality of life [37], occupational stress [38], anxiety [39], and depression (in college students) [40]. The VAS scale has also been used before in other studies involving university students [20], which makes the results more comparable.

When the questionnaire was given both at baseline and endline (methods M1 and M2), the possibility of remembering previous answers was more likely. In this case, the participants were exposed to a higher risk of answering in an inaccurate manner (e.g., giving socially desirable answers). To avoid socially desirable answers, the questionnaire was kept completely anonymous.

It is uncertain whether the short field interviews reached full data saturation. Finally, as most of the questionnaires did not have a unique identifier, it was not possible to compare baseline–endline values at an individual level but only at a group level, except for Scenario 1.

Questionnaires and interviews occurred in English. Although the master’s students had some of their courses taught in the English language, this may have been a barrier for some.

As this intervention was limited in time, there are no data related to how effective the PN breaks would be in the long term (e.g., for a whole semester) or if a novelty effect should be considered.

## 5. Conclusions and Perspectives

This pilot study was intended to inform a full study. In all but one tested scenario, increases in engagement facilitators (including alertness, concentration, enjoyment, motivation, and positive mood) attributable to the PN breaks were observed. The magnitude of these increases ranged from 0.4 to 1.8 out of 10 on a 0-to-10 VAS scale. As these preliminary results indicate a positive effect of PN, future studies could continue integrating movement breaks of less than one minute and test feasibility and effectiveness on a larger sample of groups across different faculties. Ideally, the PN breaks should be tested in courses with similar levels of interaction, e.g., traditional lectures or flipped classrooms, to improve comparability and for a prolonged time to assess the effects in the long term. The questionnaire Q2 should be preferred as a simpler and quicker measurement tool that minimizes disruption and survey fatigue.

Leading a movement break might be difficult for some lecturers; therefore, videos of people leading movement breaks that are ready to be inserted into a slideshow could be provided. The availability of such material would facilitate future expansion within the university context.

## Figures and Tables

**Figure 1 ijerph-22-00739-f001:**
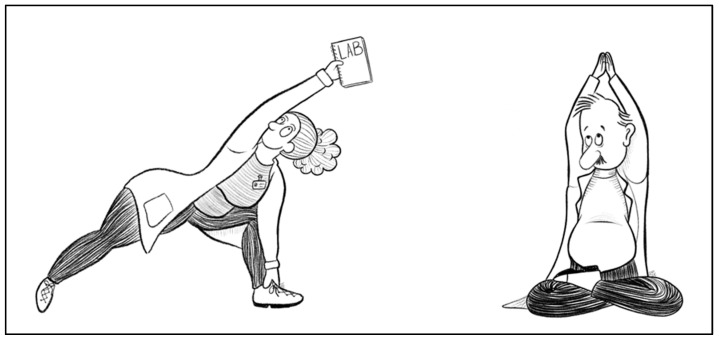
*PauseNørd* pilot study: Two examples of *PauseNørd drawings*; the humorous drawings of researchers performing physical exercises were used as a signal to begin a *PauseNørd* break. Artist: Sissel Helbæk Mogensen.

**Figure 2 ijerph-22-00739-f002:**
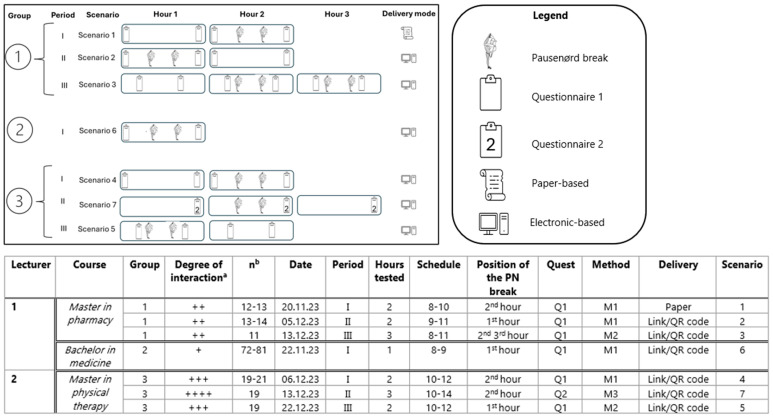
*PauseNørd* pilot study: overview of the research process of the intervention. Control lecture hours and *PauseNørd* break hours alternated in each study period. *PauseNørd* breaks occurred approximately every 15 min. ^a^ Degree of interaction based on the kind of lecture (without taking into account interactions during the PN breaks). The range of degree of interaction of the lectures was ranked from 1 to 4, with “+” indicating lower interaction (classical lecture with lecturer talking) to “++++” indicating higher interaction (lecture with group assignment work); ^b^ Number of students present in the class at each questionnaire completion time. A range occurred in case of students arriving late. Abbreviations: PN, *PauseNørd*; Quest, Questionnaire.

**Table 1 ijerph-22-00739-t001:** Overview of the primary and secondary outcomes of the *PauseNørd* pilot study on mini movement breaks.

Area of Feasibility Investigated	Method	Kind of Data	Outcome
PN breaks’ acceptability	Observation	Quantitative	Participation rate (% of total eligible) in intervention and questionnaires (s)
	Observation	Qualitative	Students’ engagement and reactions
	Interview (s/l)	Qualitative	Experiences and opinions about PN (s/l)
PN breaks’ practicality	Observation	Qualitative	Practical aspects (space, timing, frequency, and safety) (s/l)
	Interview (s/l)	Qualitative	Suggestions to improve practicality (s/l)
PN breaks’ expansion	Interview (s/l)	Qualitative	Barriers and facilitators (s/l)
PN breaks’ limited efficacy testing	Questionnaire	Quantitative	Differences in alertness, concentration, enjoyment related to the class, motivation for following the class, sleepiness, cognitive fatigue, restlessness due to prolonged sitting, social interactions within the class, and positive mood between classes with and without PN breaks (s)
Questionnaire acceptability	Interview	Qualitative	Experiences and opinions (s/l)

Abbreviations: PN, *PauseNørd*; s/l, students and lecturers.

**Table 2 ijerph-22-00739-t002:** Participation rates in PN breaks and questionnaire completion by group.

Group and Period	Date	Scenario	Intervention Hour	Students in Class	Participation Rate for the Questionnaires—n (%)	PN Breaks	Participation Rate for the PN Breaks—n (%)
** *Group 1* **							
Period I	20-11-2023	Scenario 1	Hour 1—baseline	12	12 (100)	n.a.	n.a.
			Hour 1—endline	13	13 (100)	n.a.	n.a.
			Hour 2—baseline	13	13 (100)	First PN	13 (100)
			Hour 2—endline	13	13 (100)	Second PN	13 (100)
Period II	05-12-2023	Scenario 2	Hour 3—baseline	13	12 (92.3)	First PN	14 (100)
			Hour 3—endline	14	14 (100)	Second PN	14 (100)
			Hour 4—baseline	14	13 (92.9)	n.a.	n.a.
			Hour 4—endline	14	14 (100)	n.a.	n.a.
Period III	13-12-2023	Scenario 3	Hour 5—baseline	11	8 (72.7)	n.a.	n.a.
			Hour 5—endline	11	8 (72.7)	n.a.	n.a.
			Hour 6—baseline	11	9 (81.8)	First PN	11 (100)
			Hour 6—endline	11	8 (72.7)	Second PN	11 (100)
			Hour 7—baseline	11	7 (63.6)	First PN	11 (100)
			Hour 7—endline	11	8 (72.7)	Second PN	11 (100)
** *Group 2* **							
Period I	22-11-2023	Scenario 6	Hour 1—baseline	72	69 (95.8)	First PN	70 * (97.2)
			Hour 1—endline	81	72 (88.9)	Second PN	79 * (97.5)
** *Group 3* **							
Period I	06-12-2023	Scenario 4	Hour 1—baseline	19	18 (94.7)	n.a.	n.a.
			Hour 1—endline	21	17 (81.0)	n.a.	n.a.
			Hour 2—baseline	21	19 (90.5)	First PN	20 (1 not, ill) (95.2)
			Hour 2—endline	21	19 (90.5)	Second PN	20 (1 not, ill) (95.2)
Period II	13-12-2023	Scenario 7	Hour 3 (method B)	19	12 (63.2)	n.a.	n.a.
			Hour 4 (method B)	19	15 (78.9)	First and second PN	18 * (94.7)
			Hour 5 (method B)	19	12 (63.2)	n.a.	n.a.
Period III	20-12-2023	Scenario 5	Hour 6—baseline	19	15 (78.9)	First PN	19 (100)
			Hour 6—endline	19	15 (78.9)	Second PN	19 (100)
			Hour 7—baseline	19	14 (73.7)	n.a.	n.a.
			Hour 7—endline	19	15 (78.9)	n.a.	n.a.

Values indicate n or n(%) values. * Note: Participation rate was measured by direct observation and varied between 1 and 2 students in Group 2. Abbreviations: n.a. (not applicable—control lecture hours); PN, *PauseNørd*.

**Table 3 ijerph-22-00739-t003:** Qualitative data from the short field interviews related to the feasibility areas investigated: acceptability, practicality, expansion, and limited efficacy testing.

Feasibility Area		Description	Quotes
**1. Acceptability**	Positive	Something active Something short and not disruptive Something collective Something energetic Something positive/fun/nice A mind break for both students and lecturers Easy to integrate in a lecture	*It was nice to get up, standing and do something for a short, just a short time* (Student 3, Master’s in Pharmacy). *At start I thought it was maybe a little bit hilarious, but I actually like PauseNørd (PN)* (Student 1, Master’s in Pharmacy). *The students seemed positive and engaged* (Lecturer 2). *I enjoyed it. It was a nice break and I felt there were smiles and laughter in the classroom […] No, I don’t think it was disruptive, it was too short* (Student 5, Master’s in Physiotherapy). *The physical exercise and the fact that you’re together in it, I think creates a strong distraction away from what we’re teaching. And in many other contexts just a distraction would not be very helpful but if you don’t do it too often and if it’s not too extensive either, then it’s my experience that it’s very easy for both me and the students to find the track that we have left. So in a way, I think the physical component of it is to wipe their minds, so they’re not immersed in the teaching all the subject matter. But it’s something else. Something collective, perhaps. Something collective that we are all in together* (Lecturer 1). *It’s surprisingly easy to integrate PN breaks in a lecture* (Lecturer 1).
	Negative	Disruptive if you cannot relax	*I think the concept is good to be active. But I think as well that it’s a little bit interrupting. Because this course is very hard, especially for me. So every time we have to do exercises, I was thrown out of the things we were just doing […] Personally it takes me more than one minute to relax* (Student 6, Master’s in Physiotherapy).
	Exceptions	Not for the days when you are sick	*I was sick, so when I heard of PN I guess I felt like I don’t want to be part of it, because I think it’s hard enough just to sit* (Student 6, Master’s in Physiotherapy).
**2. Practicality**	Suggestions to improve practicality	Easier to integrate without the testing time (questionnaires) More PN in the afternoon and when many hours are scheduled More PN in heavier and less interactive lectures Nice to choose your own exercise if you know what you are doing Risk of disruption if you are working at a group assignment during the lecture Address special issues before, e.g., talking privately to students in wheelchairs before introducing the PN breaks	**Questionnaire frequency** *[related to the questionnaire] I think there were too many questions* (Student 1, Master’s in Pharmacy). *I remember it [the questionnaire] was short, easy to fill out. However [if I had to fill out two questionnaires several times for several weeks] it would be too much* (Student 5, Master’s in Physiotherapy). **Time of the day, frequency, schedule, and kind of lecture** *I think it was a great idea, but I think it also depends on what course you’re in. Because some courses are very heavy, and some are not as heavy. So I prefer PN in more heavy lectures, of course […] for the longer days and the more heavy subjects* […] *today we only have three hours and then we have regular breaks in between, so I think like one [PN] in the middle of the class would be fine. […] Whenever it’s very heavy and you’re sitting for many hours and there’s no interaction, then there would be a great idea to have more PN breaks* (Student 3, Master’s in Pharmacy). *I think it works at any time* (Student 1, Master’s in Pharmacy). *I think it would be good to have it [PN] especially in the afternoon, like if you have late hours like say from three to six where most people are like very tired […] but it works also in the morning* (Student 2, Master’s in Pharmacy). *Sometimes I would like the PN, especially when we have those late hours and we have many hours in a row […] Here in the first hour [in the morning], I think it’s not necessary to have that PN […] after that I would personally need that little break* (Student 4, Master’s in Pharmacy). *I think for me at least it would always be positive* (Student 5, Master in Physiotherapy). *I think this class from the start demanded that we started on our assignment, so we also tend to use our breaks to work or process what we’ve just learned. So these hours when we’re here we’re really focused on working, so, but in other classes […] I think it will work better because you’re not working on an assignment* (Student 6, Master’s in Physiotherapy). *I think all normal lectures would work fine with the PN. […] It would maybe not work in the laboratory* (Students 1 and 2, Master’s in Pharmacy). *I think that PN could be implemented in many other settings, I don’t see any limitation* (Lecturers 1 and 2). **Kind of exercise** *I think in one of the breaks [the lecturer] said to one of the students that they could choose [which exercise to do during the PN break], and I thought that’s also a good idea, try to delegate to us the assignment. Especially because we’re physiotherapists* (Student 5, Master’s in Physiotherapy). **Special issues** *I have in the past come across students with a wheelchair, and I usually go to them before the class starts and tell them that there will be exercises during the lesson and they appreciate being informed* (Lecturer 1).
**3. Expansion**	Potential barriers	Lecturers comfortable in leading exercises Lecturer and student culture	*If we wanted to do this in a large scale at the university, there would be some barriers: not all teachers would be comfortable doing this, I suspect […] I have no anxiety in teaching, so I can do that, but I’m not sure everyone can* (Lecturer 1). *I also think this population is maybe special […] we would not be in this career path if we didn’t enjoy exercise or have positive thoughts about it; but maybe if we did it [PN] in our statistics class, where there’s also nurses and people with other people health backgrounds, I don’t think everybody would be as positive about it. Maybe over time they would get to enjoy it and feel that “Oh! It’s actually a nice break and I felt like I cleared my thoughts”* (Student 5, Master’s in Physiotherapy). *I don’t see any limitation, apart from the fact that people are not used to that* (Lecturer 2).
**4. Limited efficacy testing**	Potential benefits	Attention Mood Enjoyment Social interactions Energy Mental break Mitigation of sedentarism risks	*It’s my own clear impression was that it helped my attention and the students’ as well* […] *From where I stand I see a lot of really, really smiling faces. So there, there’s clearly something on the mood as well* (Lecturer 1). *I think it’s fun because we laugh, so it’s not just the standing and moving, but we also have fun* (Student 1). *I think it was a good small break to have in the teaching. I think it gave some energy, so I think that was good* (Student 2). *There is the need to implement it across departments and faculties, I think, because we sit for too long for too many hours* (Lecturer 2). *Yeah, it [PN] helped. I think for me, sitting down a long day, I feel like I need to move* (Student 5).
**5. Questionnaire acceptability**		Easier to integrate PN without the testing time (questionnaires)	**Questionnaire frequency** *[related to the questionnaire] I think there were too many questions* (Student 1, Master’s in Pharmacy). *I remember it [the questionnaire] was short, easy to fill out. However [if I had to fill out two questionnaires several times for several weeks] it would be too much* (Student 5, Master’s in Physiotherapy).

**Table 4 ijerph-22-00739-t004:** Limited efficacy testing: outcomes in Scenario 1—questionnaire scores for Group 1, Period I, h8 and h9: numbers of students showing increased, equal, or decreased scores, attributed to the presence of the *PauseNørd* breaks.

Variable	Increased	Equal	Declined	Missing	*p*-Value
Engagement facilitators	9	1	2	1	0.0166
Social interactions	10	1	-	2	0.0037
Engagement barriers	4	2	6	1	0.4299
Sleepiness	5	4	3	1	0.4939

n = 13. *p*-values obtained by Wilcoxon signed rank test. The *p*-values generated by a paired t-test (unequal variance) comparing the change with and without the PN breaks were 0.0029 (engagement facilitators), <0.001 (social interactions), 0.2668 (engagement barriers), and 0.6255 for sleepiness.

**Table 5 ijerph-22-00739-t005:** Limited efficacy testing: outcomes in scenarios 1–6. Comparison of changes in lectures with and without *PauseNørd* breaks by group and period.

Intervention Settings					Engagement Facilitators			Social Interactions			Engagement Barriers			Sleepiness		
Group	Date	Scenario	Position of the PN Break	Timing	Δ No	Δ Yes	ΔΔ	Δ No	Δ Yes	ΔΔ	Δ No	Δ Yes	ΔΔ	Δ No	Δ Yes	ΔΔ
1	20-11-2023	1	2nd hour	T1	−0.6	1.2	**1.8**	−2.0	1.3	**3.3**	0.1	−0.7	−0.8	−0.6	−0.5	0.1
1	05-12-2023	2	1st hour	T1	−0.4	0.6	**1.0**	−0.4	0.4	0.8	−0.1	−0.3	−0.2	−1.0	−0.5	0.5
1	13-12-2023	3	2nd and 3rd hour	T2	−0.2	0.5	0.7	0.0	1.0 ^a^	**1.0**	0.4	−0.1	−0.5	−0.8	−0.3	0.5
2	22-11-2023	6	1st hour	T1	n.a.	0.9	n.a.	n.a.	0.4	n.a.	n.a.	−0.1	n.a.	n.a.	−1.5	n.a.
3	06-12-2023	4	2nd hour	T1	0.2	−0.5	−0.3	−0.5	0.2	0.7	0.4	−0.4	−0.8	−0.0	−0.6	−0.7
3	20-12-2023	5	1st hour	T2	−0.3	0.1	0.4	−0.2	0.8	**1.0**	0.8	0.4	−0.4	0.3	−0.3	−0.5
*Avg*					−0.3	0.5	0.7	−0.6	0.7	1.4	0.3	−0.2	−0.5	−0.4	−0.6	−0.0

Abbreviations: PN, *PauseNørd*; Avg, average; Δ no (delta-no) mean difference during the control lectures without the PN breaks; Δ yes (delta-yes), mean difference during the intervention lectures with the PN breaks; ΔΔ, delta-delta, mean difference between the changes in the delta-yes and the delta-no, the change attributable to the PN breaks; n.a. not applicable (control lecture hour not assessed).

**Table 6 ijerph-22-00739-t006:** Limited efficacy testing: outcomes in Scenario 7 and comparison with the results from Peiris et al. (2021) [20].

Variable	PauseNørd ^a^ (n = 15)	No PauseNørd ^b^ (n = 24)	Mean Difference (No PauseNørd vs. PN)	Peiris et al. Movement Break (Mean, SD) Class A and Class B	Peiris et al. No Movement Break (Mean, SD) Class A and Class B	Peiris et al. Difference (Mean, 95% CI) Class A and Class B	Peiris et al. *p*-Value Class A and Class B
Alertness	6.9	5.2	1.8	7.0 (1.5) 6.6 (1.5)	5.5 (1.7) 5.2 (1.8)	1.5 (1.1 to 1.9) 1.4 (1.0 to 1.8)	<0.001 <0.001
Concentration	6.5	5.0	1.5	6.8 (1.6) 6.7 (1.6)	5.4 (1.8) 5.0 (1.8)	1.4 (1.0 to 1.9) 1.6 (1.2 to 2.1)	<0.001 <0.001
Enjoyment	7.3	6.7	0.6	7.3 (1.6) 7.0 (2.0)	5.6 (1.8) 5.5 (1.8)	1.7 (1.3 to 2.2) 1.6 (1.1 to 2.0)	<0.001 <0.001
Motivation	7.7	6.3	1.4				
Sleepiness	4.9	5.2	−0.3				
Cognitive fatigue	3.9	5.9	−2.1				
Restlessness	4.1	5.4	−1.2				
Social interactions	6.7	7.4	−0.7				
Positive mood	7.7	7.3	0.4				

Values indicate means. ^a^ Means obtained by one lecture, lecture h: 11am. ^b^ Pooled means obtained by two lectures for lectures at 10am (n = 12) and 1pm (n = 12). Notes: No missing observations. Summarizing variables (engagement facilitators and engagement barriers) were not used in this analysis to facilitate comparison with the results elaborated by Peiris et al. (2021) [20].

## Data Availability

The data presented in this study are available on request from the corresponding author due to privacy restrictions.

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
