# Peer review of "PauseNørd* Pilot Study: Exploring the Implementation of Mini Movement Breaks in University Lectures"

_ijerph, 2025, doi:10.3390/ijerph22050739_

Round 1
Reviewer 1 Report
Comments and Suggestions for Authors
The article entitled PauseNord pilot study: exploring the implementation of mini-movement breaks in university lectures applies an evidence-based practice from elementary/primary school to the university setting. The research questions are novel and interesting but the methodology has fatal flaws, therefore prohibiting its publication in IJERPH. Below are specific comments.
- While one-minute exercise breaks have not been applied in the classroom setting, their utility in other settings has been studied. See literature on "exercise snacks and recent review article published by Islam, Hashim; Gibala, Martin J.; Little, Jonathan P.. Exercise Snacks: A Novel Strategy to Improve Cardiometabolic Health. Exercise and Sport Sciences Reviews 50(1):p 31-37, January 2022. | DOI: 10.1249/JES.0000000000000275. Even though the primary outcome in this review is cardiometabolic health, this body of literature may be more appropriate given the the focus on adult populations. Developmental differences between primary and university students are notable, thereby reducing the utility of direct comparisons.
- More information is needed regarding requiring ethical consent. What is the official language given from the ethics committee that deems this research exempt? This information and the rationale for exemption status should be included.
- The Materials and Methods section is very difficult to follow. Consider reframing to directly align with the research questions being asked.
- Terms such as acceptability, practicality, expansion, and limited efficacy need to be defined and the questions/questionnaires that assessed each of these constructs should be clearly identified. It is unclear if validated questionnaires were used of new questionnaires were developed to address these questions.
- Evidence-based direct observation methodologies were not used. How was the accuracy of the direct observations determined? Were their multiple observers and if so, what was the reliability across observers. This data does not appear to be collected in a rigorous manner. Data from the primary classroom setting, use methods such as SOPLAY.
- Line 105 notes that one of the researchers would initiate the PN break when it was not feasible to put a drawing into the slideshow. This confounds the measurements of feasibility. Why didn't the lecturer initiate the PN break?
- More details are needed to describe the differences in the different type of lecture formats: flipped, classic, and group assignment. How much time is typically spent in sitting during these different lecture formats? There is also the confounding variable of class-size. How was this accounted for in your study?
- The description of the data analysis for qualitative data and the integration of data for the mixed methods is underdeveloped. More details are needed regarding the design of the mixed methods study (sequential or concurrent), software used for identifying themes, trustworthiness of data, etc.
Minor comments.
- There is quite a bit of repeated information in the Materials and Methods section. The paper should be streamlined to introduce important concepts once.
- More formal language should be used (e.g., we're vs we are/ didn't vs did not)
Reviewer 2 Report
Comments and Suggestions for Authors
I congratulate the Authors on an interesting research idea. It seems that an effective “active” break for students is extremely valuable and will support the absorption of knowledge by stimulating cognitive processes, among other things. In my opinion, the work is written very well. Both the material, methods and results are comprehensive and do not require substantive changes. I would, however, suggest a few minor comments:
- I expected a more elaborate introduction justifying the research problem. In my opinion, the introduction should be longer.
- in my opinion, the first objective of the study should be expanded and explained in more detail. The main contribution of this work should be found here.
- the quality of figure 2 should be improved.
- the second paragraph of the discussion should be more expanded and the authors should use more papers to compare the results obtained.
Reviewer 3 Report
Comments and Suggestions for Authors
Dear Authors,
I appreciate the opportunity to review your manuscript. Below are my comments and suggestions to enhance its clarity, structure, and scientific rigor.
Abstract
The abstract follows the standard format. However, I recommend reducing the number of keywords to ensure conciseness and relevance.
- Introduction
The introduction is relatively short, comprising 12 references dating from 2011 to the present. I suggest extending this section by incorporating additional background information to provide a stronger foundation for the study. I particularly appreciate the clarity and formatting of the first and second objectives.
- Line 47: Please use full forms
- Materials and Methods
This section is well-described. The total number of participants (n = 116) is appropriate for a pilot study. However, it is notable that the majority of participants were bachelor students in Medicine (n = 81), which may limit the generalizability of the findings.
- Figures 1 and 2 are well-designed and effectively illustrate the methodology.
- Table 1 is clear and includes all the necessary data.
The results section is well-structured and presents the findings clearly.
- Line 216: A full stop is missing.
- Line 240: Why did some students remain seated? A brief clarification would be helpful. I believe that all students should have performed the same exercises to ensure consistency in the analysis.
- When using abbreviations, I suggest defining them initially (e.g., physical activity (PA)) and using them consistently throughout the text.
- Tables 2–6 are well-designed and effectively present the data.
- Discussion
The discussion is well-written, providing an insightful analysis supported by adequate and relevant references. I particularly appreciate the section on Strengths and Limitations, which adds depth and transparency to the study.
- Conclusions and Perspectives
This section is well-structured; however, I recommend incorporating some statistical data to reinforce its scientific rigor. Extending the conclusions slightly would enhance their impact and provide a stronger link to future perspectives.
The manuscript is well-prepared with a clear structure and strong methodological framework. The suggested revisions are minor and do not affect the overall quality of the study. Therefore, I recommend publishing it with minor changes.
